# Population dynamics of free-roaming dogs in two European regions and implications for population control

**Lauren Margaret Smith** [1]*, **Conor Goold**[1], **Rupert J. Quinnell**[1], **Alexandru M. Munteanu**[2], **Sabine Hartmann**[2], **Paolo Dalla Villa**[3,4], **Lisa M. Collins** [1]*

**1** Faculty of Biological Sciences, University of Leeds, Leeds, United Kingdom, **2** VIER PFOTEN International, Vienna, Austria, **3** Istituto Zooprofilattico Sperimentale dell'Abruzzo e del Molise "G. Caporale", Teramo, Italy, **4** World Organization for Animal Health, OIE Sub-Regional Representation in Brussels, Brussels, Belgium

* lauren.m.smith026@gmail.com (LMS); L.Collins@leeds.ac.uk (LMC)

**Data Availability Statement:** All data, code and supporting files are available from Github: https://

## Abstract

Changes in free-roaming dog population size are important indicators of the effectiveness of dog population management. Assessing the effectiveness of different management methods also requires estimating the processes that change population size, such as the rates of recruitment into and removal from a population. This is one of the first studies to quantify the size, rates of recruitment and removal, and health and welfare status of free-roaming dog populations in Europe. We determined the size, dynamics, and health status of free-roaming dog populations in Pescara, Italy, and Lviv, Ukraine, over a 15-month study period. Both study populations had ongoing dog population management through catch-neuter-release and sheltering programmes. Average monthly apparent survival probability was 0.93 (95% CI 0.81–1.00) in Pescara and 0.93 (95% CI 0.84–0.99) in Lviv. An average of 7 dogs km$^{-2}$ were observed in Pescara and 40 dogs km$^{-2}$ in Lviv. Per capita entry probabilities varied between 0.09 and 0.20 in Pescara, and 0.12 and 0.42 in Lviv. In Lviv, detection probability was lower on weekdays (odds ratio: 0.74, 95% CI 0.53–0.96) and higher on market days (odds ratio: 2.58, 95% CI 1.28–4.14), and apparent survival probability was lower in males (odds ratio: 0.25, 95% CI 0.03–0.59). Few juveniles were observed in the study populations, indicating that recruitment may be occurring by movement between dog subpopulations (e.g. from local owned or neighbouring free-roaming dog populations), with important consequences for population control. This study provides important data for planning effective dog population management and for informing population and infectious disease modelling.

## Introduction

Domestic dogs (*Canis familiaris*) are abundant globally, with the total population size estimated at around 700 million to 1 billion [1,2]. Dogs that are unrestricted in their movement, without human supervision, are part of the free-roaming dog population. This includes both owned and unowned dogs. The free-roaming dog population can present issues in terms of

github.com/lauren-smith-r/Smith-et-al-Population-dynamics-free-roaming-dogs.

**Funding:** LMC has received a research grant from VIER PFOTEN International (https://www.four-paws.org); and LMS's research has been funded by VIER PFOTEN International. SH and AMM are employed by VIER PFOTEN International and contributed to the conceptualisation of the study and reviewing and editing of drafts.

**Competing interests:** The authors declare that: A. M.M. and S.H. are employed by VIER PFOTEN International, a global animal welfare organisation; L.M.C has received a research grant from VIER PFOTEN International; and L.M.S.'s research has been funded by VIER PFOTEN International. This does not alter our adherence to PLOS ONE policies on sharing data and materials.

public health [3–5], conservation of wildlife [6], livestock predation [7,8], and dog welfare [9,10]. Effective management of the free-roaming dog population is a primary concern of government agencies, animal welfare organisations, and public health and conservation researchers [11]. Changes in free-roaming dog population size are important indicators of the effectiveness of dog population management. Reducing population size and stabilising population turnover can lead to reductions in risks to public health [12,13], conservation of wildlife [6], and dog welfare [9,14,15].

Dog population size is a function of the processes of recruitment and removal, such as births, deaths, immigration, and emigration. Several studies have estimated rates of overall recruitment and removal, births, mortality, migration and dispersal of free-roaming dog populations [10,16–18]. Studies mostly use household questionnaires and/or direct observation of dog populations to attain estimates. Questionnaires take advantage of the loose ownership status of free-roaming dogs and allow monitoring of individuals over several years through repeated surveys [16–18]. While questionnaires may be applicable for populations where free-roaming dogs are mostly owned, they may preferentially sample those dogs under stricter controls than dogs that are unclaimed or unrestricted in their movements. Most free-roaming dogs have been reported as owned in some respect [16,19–21], though previous studies have largely been conducted in Asia, Africa and South America, with a lack of data from dog populations in Europe.

The few studies that have been carried out in Europe have been limited to estimating populations sizes of free-roaming (e.g. in Poland [22]) or owned dogs (e.g. in Italy [23,24]). No studies have estimated rates of recruitment or removal in European free-roaming dog populations. Dog population management in Europe is conducted to reduce risks to public health (e.g. rabies and leishmaniasis) [25–28], reduce predation on livestock [8] and wildlife [29], and to improve free-roaming dog welfare [30]. As dog population dynamics are likely to vary between countries, relating to the habitat type (e.g. urban/rural) and the human population (e.g. density and cultural/social factors) [1,9], it is important that we better understand the dynamics of free-roaming dog populations in Europe to help inform management strategies.

Mark-recapture is a commonly used method of estimating population size, where individuals are observed, identified (e.g. through marking), and re-observed during successive surveys to calculate abundance and population processes. Dog population size has often been estimated using closed mark-recapture methods (see [31] for review) that assume geographic and demographic closure (i.e. no births, deaths or migration). Closed mark-recapture methods allow the estimation of population size and detection probability [32]. They are advantageous as they can allow for individual heterogeneity in detection probability and differences in detection probability after first capture, leading to less biased parameter estimates [32]. Closed mark-recapture methods do not estimate recruitment (i.e. births, immigration, and abandonment of dogs) and removal (i.e. deaths, emigration, and adoption of dogs) rates, which describe how the population is changing. Open mark-recapture methods account for these demographic and geographic processes and allow estimation of recruitment and removal rates (e.g. Jolly-Seber, Cormack-Jolly-Seber, and Pollock's robust design).

To reduce population size, population management methods aim to alter rates of recruitment or removal in a population. For example, increasing removal of individuals through culling or sheltering, and decreasing recruitment by reducing births. Open mark-recapture methods are useful for understanding how the population is changing through the relative rates of recruitment and removal, and for assessing the potential effectiveness of different management methods, for example through elasticity or perturbation analyses [33]. Open mark-recapture methods have rarely been applied to free-roaming dog populations [31,34], possibly due to a lack of awareness, expertise, or resources to apply open mark-recapture methods. Belo

et al., (2017) [34] are the first to report dog demographic parameter estimates through an open mark-recapture (Jolly-Seber) approach in Brazil. Belo et al., (2017) [34] report high population turnover (high removal and recruitment rates), which has implications for population and disease control. They attributed the high recruitment rates to the abandonment of owned dogs, suggesting that methods reducing owned dog abandonment will be most effective in this area [34].

The Pollock's robust design method is advantageous over other methods as it incorporates both open and closed mark-recapture study designs and analyses [35,36]. The robust design is a nested sampling design incorporating sampling occasions over two temporal scales, involving widely spaced primary sampling periods, where the population is assumed open to the influences of recruitment and removal, and narrowly spaced secondary sampling periods, where the population is assumed closed to the influences of recruitment and removal. By incorporating both methods, Pollock's robust design allows for the demographic processes of recruitment and removal to be estimated (as in open models), and also deals with individual heterogeneity in detection probability (as in closed models) and survival probability [35,36], providing more robust parameter estimates. Whilst the Pollock's robust design mark-recapture method has been applied to a number of animal populations [37–39], it has not previously been applied to free-roaming dogs.

The aim of this study was to determine the size, dynamics, and health status of two European free-roaming dog populations. We investigate these parameters using two free-roaming dog populations in Pescara, Italy and Lviv, Ukraine. In central Italy, free-roaming dog populations contribute to the spread of leishmaniasis (see [40,41] for review) and predation of livestock [8]. Rabies is a public health concern in Ukraine, which has the second highest incidence of rabies cases in Europe [42]. Dog population management is conducted in both study regions to control the population size and the risks associated with the free-roaming dogs populations [11,30]. Understanding the free-roaming dog population dynamics in the study regions primarily helps to inform management strategies within Italy and Ukraine, although the results may be applicable in other regions with similar environments and human populations.

In Lviv, as different study sites had varying intensity of population management, we also aimed to investigate whether there were changes in population dynamics between sites where different dog population management had been applied. The effects of environmental factors, day of the week, and sex on detection and apparent survival probability were also investigated. Using this information, we discuss how population processes could inform dog population management.

## Materials and methods

### Study regions and study sites

This study was carried out in two regions, Pescara province in Italy, and the Lviv region of Ukraine (Fig 1). Pescara is located in central Italy in the Abruzzo region and has an oceanic climate [43]. Lviv is located in the west of Ukraine and has a temperate continental climate [43]. Regions were selected due to historical records on dog population management from the Veterinary Services—Pescara Province Local Health Unit and local Communal Enterprise in Lviv. Both regions had ongoing dog population management through a combination of catch-neuter-release (CNR) and sheltering. Data was available for 42 out of 46 municipalities in Pescara and for the entire city of Lviv. Four areas in each study region were selected to have similar: (i) number of inhabitants in each town/suburb; and (ii) profiles in terms of size, structure (e.g. residential/industrial), and household numbers (assessed visually prior to fieldwork).

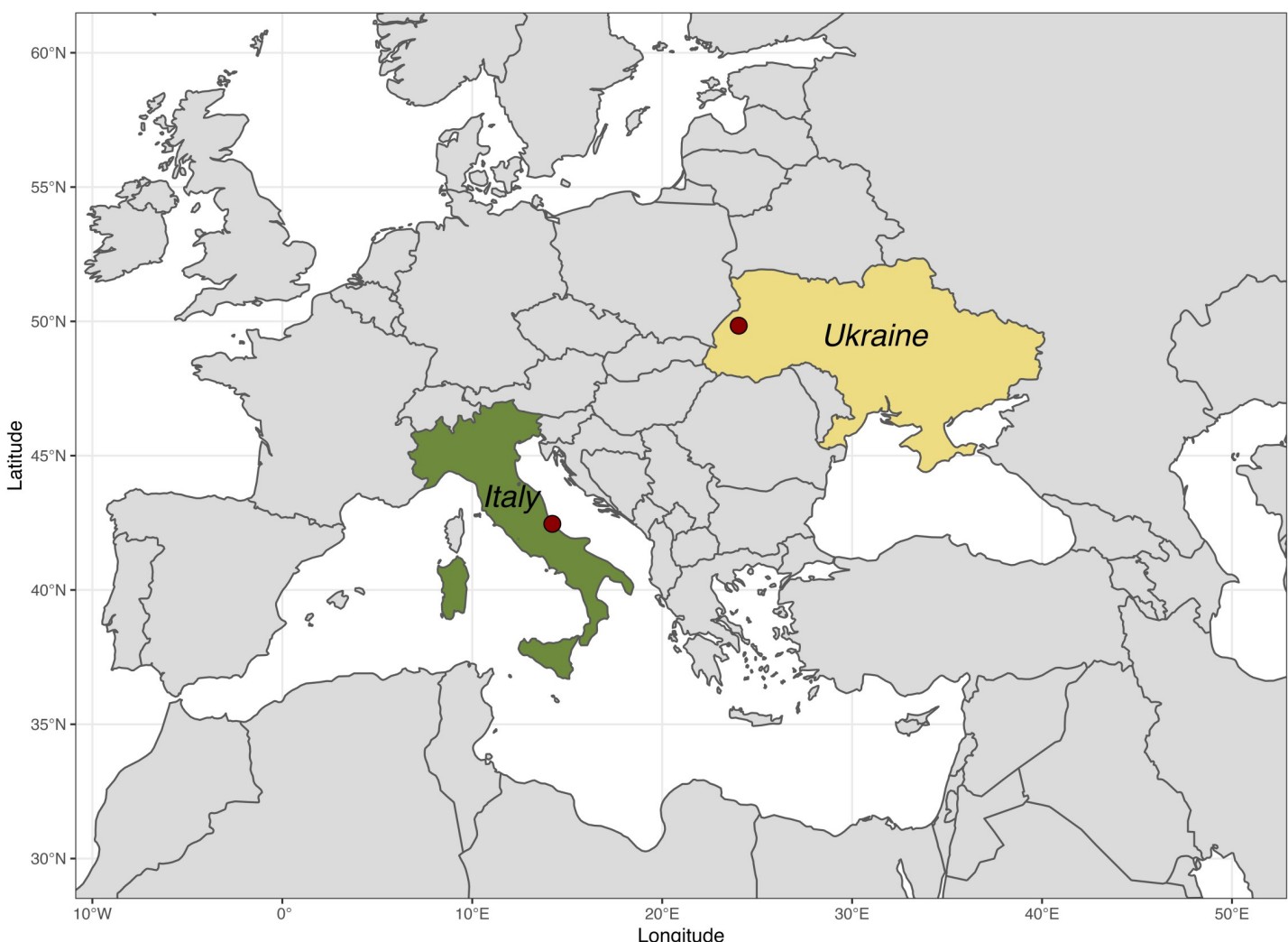

**Fig 1. Map highlighting Italy (green) and Ukraine (yellow) with study regions of Pescara and Lviv indicated by red circle.**

Prior to the fieldwork commencing, pilot trips to the study sites were conducted to check the suitability of the selected study sites for: (i) accessibility (i.e., no private land such as industrial areas where access is prohibited); and (ii) the presence of free-roaming dogs. Data collected during the pilot trip was not included in the analysis. The precise study sites remain anonymous as a condition of data sharing with the local networks. All data was collected in public areas of Pescara and Lviv (i.e. publicly accessible streets), therefore no permits were required.

In Pescara, a study site refers to a rural town/village in the Pescara province. Population density in the study sites in Pescara varied between 127 and 193 people km$^{-2}$. Distances between sites varied between 4.65 and 12.40 km. Study sites in Pescara were selected to have similar dog population management: similar numbers of dogs had been caught, neutered, and released within the study sites between 2015 and 2019 (S1 Table). In Lviv, a study site refers to a section of Lviv city. Lviv city is an urban environment with a population size of 717,803 and a population density of 3,982 people km$^{-2}$. Distances between study sites varied between 1.00 and 6.80 km in Lviv. All study sites were approximately 2 km$^2$. In Lviv, as the level of dog

population management differed throughout the city, we aimed to assess whether there were differences between sites with varying management intensity by selecting two study sites where dogs had been caught, neutered, and released (sites one and two) and two study sites where no dogs had been caught, neutered, and released (sites three and four) (S1 Table).

### Data collection

Data was collected in each study site every three months between April 2018 and July 2019 (Fig 2), excluding in January 2019, where data collection did not occur due to the logistical challenges associated with the extremely low temperatures in both study regions. Within each primary sampling period, data was collected over three consecutive days (secondary sampling periods) in each study site (Fig 2), except for in study site one in Lviv in primary sampling period three (October 2018), where secondary sampling period two was missed due to field-worker illness. As data for this day is "missing completely at random", this does not impact the results.

Data was collected using a street survey approach between approximately 7am and 9am (see S1 File and S2 Table for details) to reduce temporal variation in detection probability [44–47]. Two fieldworkers travelled together on foot along predesigned routes and recorded information on every visible free-roaming dog (Table 1). Survey routes were designed to maximise street coverage across the study site and avoid enclosed areas as a safety measure to reduce the risk of dog attack. Roads without a pavement were excluded as a traffic safety measure. The street surveys followed the same route across both the secondary and primary sampling days. Dogs were classified as free-roaming if they were: (i) not within an enclosed private property (e.g. the front yard of a house); (ii) not on a lead; and (iii) not associated with a person (i.e. not on a lead but under the watch and responsibility of a person). All information was logged on the Animal-id.info app (animal-id.net); a mobile software application. This app facilitates data collection by storing Global Positioning System (GPS) coordinates and allows easy logging of all dog demographic and health variables (Table 1). To reduce inter-observer variation, field-workers undertook training prior to fieldwork on how to score the body condition of dogs. Fieldworkers photographed every observed dog using a Nikon D3400 camera for subsequent

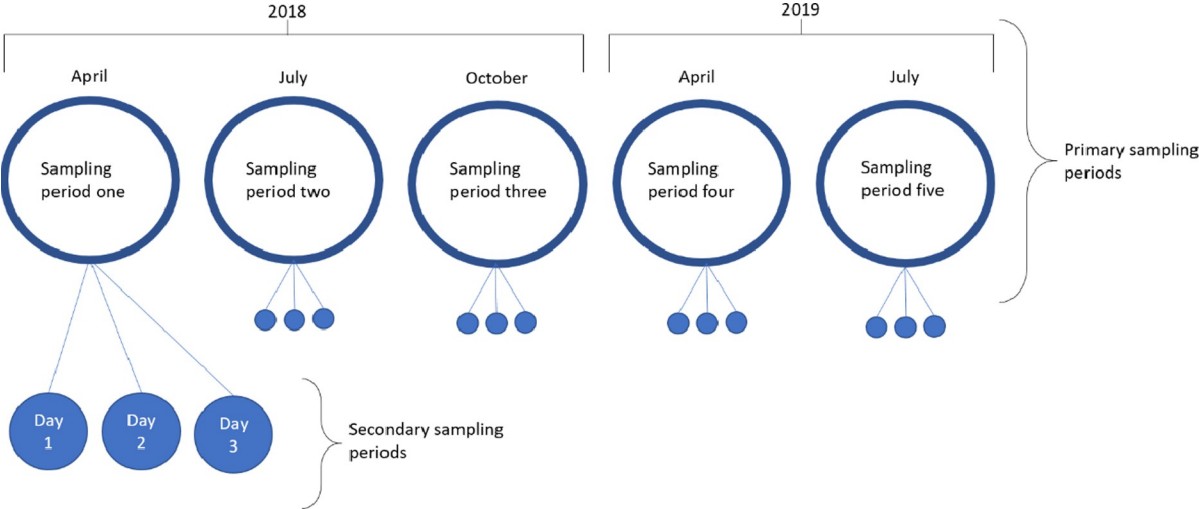

**Fig 2. Study design consisting of five primary sampling periods conducted at three-month intervals between April 2018 and July 2019 (excluding January 2019) and three consecutive days of secondary sampling periods within each primary sampling period.**

**Table 1. Variables recorded during street surveys.**

| Variable | Categories | Method of estimation |
|---|---|---|
| Global Positioning System (GPS) coordinates | Latitude and longitude | GPS recording in Animal-id.info app |
| Sex | Male / female / unknown | Observation of reproductive organs |
| Age | Juvenile (less than one year) / adult (over one year) | Body size, allometry and behaviour [47] |
| Size (height) | Large (>65cm) / medium (45-65cm) / small (<45cm) | Estimated visually |
| Neutering status (Lviv only) | Presence/absence | Observation of ear tag |
| Collar | Presence/absence | Observation of collar |
| Visibly pregnant (females only) | Yes/no | Observation of enlarged abdomen and mammary glands |
| Lactation status (females only) | Yes/no | Observation of enlarged mammary glands |
| Skin condition | Presence/absence | Observation of hair loss and/or dermatitis |
| Visible injury | Presence/absence | Observation of visible lesions (e.g. wounds) or lameness |
| Body condition score (non-pregnant and non-lactating adult dogs only) | 1 emaciated / 2 underweight / 3 normal / 4 overweight / or 5 obese | Based on visible body fat coverage [48] |
| Temperature at beginning and end of survey | Degrees Celsius (°C) | weather.com |
| Rain | Yes/no | Observation |
| Market during survey | Yes/no | Observation |

identification of individuals. Photographs were taken to include details of both sides of the dog's body, its legs, head, and tail.

This study was conducted by observing and photographing free-roaming dogs from a distance in public areas of Pescara, Italy and Lviv, Ukraine. All data were collected visually, i.e. no dogs were handled during this study. This study therefore did not require formal ethical approval as the study did not involve handling, husbandry or established veterinary practice, and did not include experimental practices which may cause pain, suffering, distress or lasting harm [49–51].

## Mark-recapture analysis

Individual capture histories were based on prior observations of the individual dogs during the primary and secondary sampling periods (1 = observed, 0 = not observed). Dogs were identified from the photographs, using distinctive markings on the body, legs, head, and tail. Each dog was given a distinctiveness rating between one and three (1 = very distinct, with unique colouring/marking; 2 = moderately distinct, with some identifiable colouring/marking; 3 = indistinct, mono-coloured with minimal markings (Fig 3). All individuals were included in the mark-recapture analysis, regardless of their distinctiveness rating. Observations of dogs where photograph quality was poor were not included in the mark-recapture analysis.

It is challenging to estimate demographic parameters using mark-recapture data because several ecological processes can lead to the mark-recapture histories that are observed. For example, individuals may be present in the population, but not detected during surveys, meaning their presence or absence is not an accurate estimate of whether an individual is contributing to the population processes. To deal with these challenges, a hierarchical Bayesian hidden Markov model of Pollock's closed robust design was used to analyse the mark-recapture histories for both regional locations. Hidden Markov models deal with these challenges as they allow the underlying latent states of dogs (e.g. their presence or absence in the population) to be estimated depending on observations during the mark-recapture surveys (i.e. their capture histories).

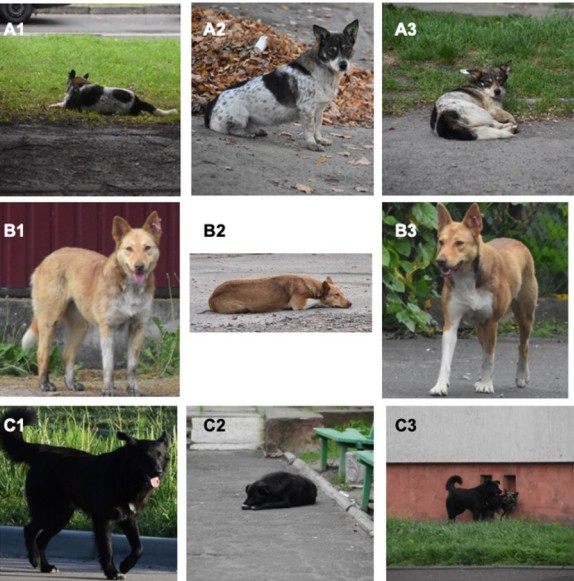

**Fig 3.** Examples of distinctiveness ratings of dogs identified across primary sampling periods: A1-3 of distinctiveness 1 (distinct with unique markings); B1-3 of distinctiveness 2 (moderately distinct, with some identifiable colouring/ markings); and C1-3 of distinctiveness 3 (indistinct, mono-coloured, minimal markings).

We used a nested study design, comprising $t$ primary sampling periods and $s$ secondary sampling periods where an individual was able to be observed. Each study site had a population of dogs that underwent the processes of recruitment (individuals entering the population through births and immigration) and removal (individuals leaving the population through death and emigration) between $t$ primary sampling periods. Between the $s$ secondary sampling periods the population was assumed closed to the processes of recruitment and removal. As described above, there were five $t$ primary sampling periods, each with three $s$ secondary sampling periods (Fig 2).

A hierarchical Bayesian hidden Markov model of Pollock's robust design was used to analyse the mark-recapture histories for both regional locations. At each primary sampling period $t$, the model estimated each dog's probability of being in one of the following three latent states, conditional on their capture history (i.e. their true states cannot be measured directly): *not-yet-entered; dead;* or *alive*. The *not-yet-entered* state described those individuals who were yet to enter the study population (i.e. through immigration or birth). The *dead* state described those individuals who were no longer part of the study population (i.e. removed through mortality, adoption to the restricted owned dog population, or emigration), and the *alive* state denoted individuals who were part of the study population. Only individuals in state *alive* were available to be observed. An individual could transition between latent states across primary sampling periods ($t$ to $t^{+1}$). No state transitions occurred between secondary sampling periods. Table 2 outlines the probability of an individual transitioning between states (*not-yet-entered, dead,* or *alive*) at primary period ($t$), given their state at the previous primary sampling period ($t^{-1}$). A dog's probability of being observed was $\delta_{t,s}$ if in the state *alive*, and zero if *dead* or *not-yet-entered*. A dog's probability of being unobserved was one if *dead* or *not-yet-entered, and* $1-\delta_{t,s}$ if *alive*.

As described by Royle & Dozario (2008) [52] and Kery & Schaub (2011) [53], full capture histories were modelled through parameter-expansion and data augmentation, which involves

**Table 2. State transition matrix: the probability of an individual transitioning to a state at primary period (t), given their state at the previous primary sampling period (t$^{-1}$) (reading from row to column).**

| | | $t^{+1}$ | | |
|---|---|---|---|---|
| | | *Not yet entered* | *Dead* | *Alive* |
| t | *Not yet entered* | 1-$\psi$ | 0 | $\psi$ |
| | *Dead* | 0 | 1 | 0 |
| | *Alive* | 0 | 1-$\varphi$ | $\varphi$ |

adding a large number of zero capture histories to account for those dogs that were present but never observed (see S2 File and S1–S5 Figs for details).

In this model, recruitment referred to the probability of an individual dog transitioning from the *not-yet-entered* state at $t^{-1}$ to *alive* at *t* primary sampling period (i.e. entering the population through immigration or birth). As discussed by Roye and Dorazio (2008) [52] and Kery and Schaub (2011) [53], recruitment probability was in practice a nuisance parameter because birth and immigration were confounded and, because of data augmentation, the recruitment probability was in fact a 'removal entry probability' that described the probability of a member of the augmented data set entering at time *t*. Removal entry probabilities thus have no biological meaning [52–54]. Instead, we followed Royle & Dozario (2008) [52] and Kery & Schaub (2011) [53] by deriving the 'entry probability' for each time point *t*, defined as the fraction of real individuals in the augmented data set that entered at each time point (S2 File). We calculated per capita probability, describing the fraction of new recruits at primary period *t* per individual dog alive and in the study site at primary period *t*. Calculation of per capita entry probabilities are detailed in S2 File. We estimated survival probability, which was also a confounded variable, as it was a function of both the probability of an individual remaining in the study site (i.e. not emigrating) and remaining alive, leading to its more popular name of "apparent survival". Table 3 outlines the parameters calculated for each study site.

All model parameters had 'weakly informative' prior distributions and all individuals started in the *not-yet-entered* state. The model was written in Stan [55] and run in R version 3.6.1 [56] using the "**Rstan**" package version 2.21.3 [57] with four Markov chain Monte Carlo chains of 2,500 iterations of warmup and 2,500 iterations for sampling, giving 10,000 posterior samples for inference. The Stan model used the forward algorithm to marginalise out the latent, discrete states for each individual. Convergence was assessed by inspecting the Rhat values (values less than 1.05 suggest convergence) and effective sample sizes (values over 1000 suggest good precision of the tails of distribution).

Random intercepts were included for *apparent survival*, *recruitment*, and *detection* to describe intra-country variation across study sites and primary periods, and intra-site variation across dogs. Spatial correlations (correlations in parameter estimates given the distances between study sites) and temporal correlations (correlations in parameter estimates given the time differences between primary sampling periods) were captured by using Gaussian process prior distributions on the sites and primary periods random intercepts (squared exponential and periodic kernel functions, respectively; see model code in https://github.com/lauren-smith-r/Smith-et-al-Population-dynamics-free-roaming-dogs). Due to the hierarchical structure, survival, recruitment, and detection results were partially pooled across dogs, sites, and primary period times.

Parameter estimates were converted from the log odds scale to the probability scale using the inverse logit function: $logit^{(-1)}(x) = {}^{\exp(x)}/_{(1+\exp(x))}$, where *x* is the posterior value on the

**Table 3. Description of parameters calculated for each study site in study regions.**

| Parameter | Description |
|---|---|
| $Z^{(m \times t)}$ | Matrix of the possible latent states (*not-yet-entered; alive; dead*) for each individual (including *pseudo-individuals*) at each *t* primary sampling period. |
| $n$ | Total number of dogs individually identified throughout the duration of the study. |
| $N_t$ | Total number of dogs alive and available for observation during primary sampling period *t*. |
| $m$ | Total number of dogs, including observed and unobserved *pseudo-individuals*. |
| $\gamma^{(m \times t \times s)}$ | Array of capture histories for all individually identified dogs and the parameter expanded data augmented *pseudo-individuals*. |
| $\gamma^{(i \times t \times s)}$ | Array of capture histories for all individuals *observed* in *s* secondary sampling periods throughout *t* primary sampling periods. |
| $W$ | Superpopulation: Total number of dogs that have ever been in the study site across all primary sampling periods. |
| $\varphi_{ti}$ | Apparent survival of individual dog between *t* and $t^{+1}$ primary sampling period. |
| $\delta_{ti}$ | Probability of observing a dog, given it is alive, in secondary sampling period *s* within primary sampling period *t*. |
| $\psi_{ti}$ | Probability of recruitment–an individual dog transitioning from *not yet entered* at $t^{-1}$ to *alive* at *t* primary sampling period. As described, this is a nuisance parameter that is required to describe the model. |
| $E_{ti}$ | Proportion of superpopulation entering at each primary period *t*, given they have not already entered. |
| $f_t$ | Per capita entry probability: the fraction of new recruits at primary period *t* per individual dog alive and in the study site at primary period *t*. |
| $\lambda$ | Population growth (S2 File). |
| $M_t$ | Matrix of time intervals between each primary sampling period. |
| $M_d$ | Matrix of distances between study sites. |

logit scale. Parameter estimates were summarised by calculating the mean and 95% credible intervals (CIs) of the posterior distribution (the 95% most probable values).

The effects of the following predictor variables were tested on detection probability: average temperature (average of recorded temperature at beginning and end of survey); market event (yes/no); rain (yes/no); weekday/weekend; sex (male/female), study site and primary period. The effects of sex, study site, and primary period were also tested on apparent survival probability. The detection and apparent survival probability for individuals of unknown sex (including pseudo-individuals) were computed by marginalising over the respective male and female conditional probabilities. For the missing survey in study site one in Lviv for primary period three, secondary sampling period two, NA's were included in the array of capture histories ($\gamma^{(i \times t \times s)}$) and, for the predictor variables, temperature, and rainfall (no rainfall) was recorded using records in weather.com, the missed survey day was a weekday and market event was recorded as NA. A significant effect was determined if the 95% CIs did not include zero on the log odds scale.

It is common in frequentist mark-recapture modelling to run several models, stratified by temporal and population subgroup parameter estimates (leading to an enormous number of possible models, see [58] for discussion) and to use model-averaging to provide parameter results. Studies suggest the use of Hierarchical Bayesian mark-recapture models with random-effects yield similar parameter estimates to model-averaging of frequentist mark-recapture models (using Akaike Information Criterion, AICc weights) [58–60]. As all parameters in the model are of theoretical relevance and as there were no specific biological hypotheses, no explicit model comparison was run.

# Results

## Descriptive statistics–dog demographics and health

In Pescara 53 dogs and in Lviv 182 dogs were individually identified. In each study region in total: an average of 10 dogs were observed in each secondary sampling period in Pescara (min = 4, max = 15), and 28 in Lviv (min = 14, max = 36); an average of 3 new dogs were identified (min = 0, max = 14) in Pescara, and 11 (min = 5, max = 36) in Lviv; and an average of 6 dogs had been previously observed (min = 0, max = 13) in Pescara, and 16 in Lviv (min = 0, max = 28) (Table 4). No individuals were observed in more than one study site in either Pescara or Lviv (i.e. there was no evidence for movement between study sites). Sex ratio was similar in both regions: of the total number of identified dogs, 22–26% were female, 51–52% were male, and 23–26% were of unknown sex (Table 5). Nearly all observed dogs were adults in both Pescara (98%) and Lviv (95%). No visibly pregnant females and few lactating females were observed in any site across both regions. All survey timings, weather, market events, and temperature conditions are detailed in S3 and S4 Tables.

**Table 4. Number of observed dogs and newly observed individuals in total across the study sites in Pescara and Lviv for each secondary sampling period.**

| Study region | Primary sampling period | Secondary sampling period | Number of dogs observed | Number of new individuals | Number of previously identified individuals |
|---|---|---|---|---|---|
| Pescara | 1 (Apr 2018) | 1 | 14 | 14 | 0 |
| | | 2 | 11 | 7 | 4 |
| | | 3 | 15 | 9 | 6 |
| | 2 (Jul 2018) | 1 | 6 | 4 | 2 |
| | | 2 | 13 | 3 | 10 |
| | | 3 | 7 | 0 | 7 |
| | 3 (Oct 2018) | 1 | 5 | 1 | 4 |
| | | 2 | 12 | 3 | 9 |
| | | 3 | 12 | 2 | 10 |
| | 4 (Apr 2019) | 1 | 8 | 1 | 7 |
| | | 2 | 10 | 3 | 7 |
| | | 3 | 4 | 0 | 4 |
| | 5 (Jul 2019) | 1 | 13 | 0 | 13 |
| | | 2 | 4 | 5 | 2 |
| | | 3 | 9 | 0 | 9 |
| Lviv | 1 (Apr 2018) | 1 | 36 | 36 | 0 |
| | | 2 | 19 | 10 | 8 |
| | | 3 | 19 | 9 | 10 |
| | 2 (Jul 2018) | 1 | 32 | 16 | 16 |
| | | 2 | 27 | 12 | 15 |
| | | 3 | 31 | 13 | 18 |
| | 3 (Oct 2018) | 1 | 31 | 13 | 18 |
| | | 2 | 27 | 7 | 20 |
| | | 3 | 35 | 10 | 25 |
| | 4 (Apr 2019) | 1 | 36 | 12 | 24 |
| | | 2 | 23 | 11 | 12 |
| | | 3 | 34 | 11 | 23 |
| | 5 (Jul 2019) | 1 | 19 | 7 | 12 |
| | | 2 | 14 | 6 | 8 |
| | | 3 | 33 | 5 | 28 |

**Table 5. Demographic and health results for observed dogs in Pescara, Italy and Lviv, Ukraine during surveys between April 2018 and July 2019.**

| | | | Pescara | Lviv |
|---|---|---|---|---|
| **Demographic** (% of total individuals) | No. individual dogs | | 53 | 182 |
| | Estimated average dog density (dogs km$^{-2}$) | | 7 | 40 |
| | Sex | Female | 26% | 22% |
| | | Male | 51% | 52% |
| | | Unknown | 23% | 26% |
| | Age | Adult | 98% | 95% |
| | | Juvenile | 2% | 5% |
| | Visibly pregnant females | | 0% | 0% |
| | Lactating females | | 7% | 5% |
| | Distinctiveness | 1 | 26% | 14% |
| | | 2 | 62% | 68% |
| | | 3 | 11% | 17% |
| **Health** (% of total observations) | Prevalence of | skin conditions | 7% | 3% |
| | | visible injuries | 12% | 7% |
| | Body condition score | 1 –emaciated | 0% | 0% |
| | | 2 –underweight | 0% | 1% |
| | | 3 –normal | 73% | 73% |
| | | 4 –overweight | 13% | 13% |
| | | 5 –obese | 3% | 2% |
| | | Unknown | 9% | 11% |
| | Neutering coverage | | NA | 34% |
| **Population dynamics** | Removal probability | | 7% | 7% |
| | Recruitment probability | | 9–20% | 12–42% |
| | Dog detection probability | | 27% | 18% |

As the different study sites in Lviv had different management strategies, the demographic and health measures between sites could be compared. Juveniles were observed in one of the two sites with CNR–site two (5 of 35 dogs, 14%)–and in both sites with no CNR–three (2 of 56 dogs, 4%), and four (2 of 64 dogs, 3%). Based on the presence of ear tags, study site one had a higher percentage of dogs neutered and vaccinated (52%), compared to study site two (29%) (Table 6). Dogs in sites three (29%) and four (17%) were also observed with ear tags, even though no/few dogs were recorded as caught, neutered, and released at these sites.

In Pescara, the overall prevalence of skin conditions was 7%, with a maximum of 21% in October 2018, while the prevalence of skin conditions was only 3% in Lviv (Table 5). The prevalence of visible injuries was higher in Pescara (12%) compared to Lviv (7%). Most dogs in both regions had a body condition score of three (normal body condition, 73% of total observations), with few observed underweight dogs.

**Table 6. Number and percentages of neutered and vaccinated dogs observed in each study site in Lviv, Ukraine.** Neuter and vaccination status indicated by presence of ear-tag. Sites with active CNR are indicated in bold.

| Study site | No. identified dogs | Neutered & vaccinated | Females neutered & vaccinated | Males neutered & vaccinated | Unknown sex neutered & vaccinated |
|---|---|---|---|---|---|
| **1** | **27** | **14 (52%)** | **7 (26%)** | **6 (22%)** | **1 (4%)** |
| **2** | **35** | **10 (29%)** | **4 (11%)** | **4 (11%)** | **1 (3%)** |
| 3 | 56 | 16 (29%) | 6 (11%) | 6 (11%) | 4 (7%) |
| 4 | 64 | 11 (17%) | 2 (3%) | 6 (9%) | 3 (5%) |

## Dog demographic parameters

The average monthly probability of a dog remaining alive and in the study population (i.e. not emigrating) was 0.93 (95% CI: 0.81–1.00) in Pescara and 0.93 (95% CI: 0.84–0.99) in Lviv. The average apparent survival probability between primary sampling periods (3 to 6 months) was 0.71 (95% CI: 0.42 to 0.95) in Pescara and 0.73 (95% CI: 0.45 to 0.95) in Lviv. The average probability of a dog being observed in a single survey (detection probability) was 0.27 (95% CI: 0.05–0.54) in Pescara and 0.18 (95% CI: 0.02–0.40) in Lviv. S5 and S6 Tables outline the apparent survival and detection probabilities per primary period and study site for Pescara and Lviv respectively. Standard deviations for between-dog effects on survival and detection are presented in S7 Table.

Per capita entry probabilities (i.e. the average fraction of dogs entering the study areas during the study periods per individual dog) varied between 0.09 (95% CI: 0.03–0.33) and 0.20 (95% CI: 0.00–0.38) in Pescara, and 0.12 (95% CI: 0.00–0.26) and 0.42 (95% CI: 0.26–0.62) in Lviv (Table 7). The average monthly per capita entry probabilities were 0.05 (95% CI 0.00–0.09) in Pescara and 0.06 (95% CI 0.01–0.10) in Lviv. Population size estimates varied between 12 (95% CI: 4–20) and 22 (95% CI: 4–41) in Pescara, and 58 (95% CI: 15–155) and 114 (95% CI: 44–195) in Lviv (Fig 4). Study sites in Pescara had an average of 7 dogs km$^{-2}$ (95% CI: 2–14 dogs km$^{-2}$) and in Lviv 40 dogs km$^{-2}$ (95% CI: 13–73 dogs km$^{-2}$) across sites and primary periods. The population in Pescara shows a declining trend throughout the study in all four study sites (Table 7 and Figs 4 and S6). In Lviv, the population increased between October 2018 and April 2019 in all four study sites (Table 7 and Figs 4 and S7). Per capita entry probabilities (Table 7) and apparent survival probabilities (S6 Table) were both high between these primary periods.

**Table 7. Estimated population size and per capita entry probability, and the 2.5 and 97.5 percentiles of the posterior distribution (95% CI,) across study sites and primary periods for Pescara, Italy and Lviv, Ukraine.** Per capita entry probabilities for the first primary period are not included, due to lack of interpretability in the primary period one parameter estimate.

| | | Estimated population size | | | | | | Per capita entry probability | | | | | |
| | | Pescara | | | Lviv | | | Pescara | | | Lviv | | |
| | Primary Period | Mean | 2.5% CI | 97.5%CI | Mean | 2.5% CI | 97.5% CI | Mean | 2.5% CI | 97.5% CI | Mean | 2.5% CI | 97.5% CI |
|---|---|---|---|---|---|---|---|---|---|---|---|---|---|
| Site 1 | 1 | 15 | 5 | 24 | 73 | 9 | 172 | | | | | | |
| | 2 | 15 | 5 | 23 | 70 | 13 | 160 | 0.20 | 0.00 | 0.38 | 0.18 | 0.00 | 0.33 |
| | 3 | 14 | 5 | 22 | 62 | 8 | 139 | 0.15 | 0.00 | 0.31 | 0.12 | 0.00 | 0.26 |
| | 4 | 12 | 4 | 20 | 81 | 18 | 168 | 0.12 | 0.00 | 0.24 | 0.33 | 0.18 | 0.55 |
| | 5 | 12 | 4 | 20 | 75 | 15 | 155 | 0.14 | 0.00 | 0.27 | 0.16 | 0.00 | 0.33 |
| Site 2 | 1 | 18 | 4 | 36 | 69 | 15 | 147 | | | | | | |
| | 2 | 17 | 4 | 33 | 66 | 18 | 133 | 0.19 | 0.00 | 0.37 | 0.18 | 0.00 | 0.34 |
| | 3 | 15 | 3 | 29 | 58 | 15 | 155 | 0.17 | 0.00 | 0.33 | 0.13 | 0.00 | 0.19 |
| | 4 | 13 | 2 | 26 | 78 | 26 | 149 | 0.12 | 0.00 | 0.26 | 0.36 | 0.19 | 0.54 |
| | 5 | 12 | 1 | 25 | 72 | 21 | 138 | 0.15 | 0.00 | 0.29 | 0.18 | 0.02 | 0.35 |
| Site 3 | 1 | 22 | 5 | 41 | 114 | 44 | 195 | | | | | | |
| | 2 | 21 | 5 | 39 | 100 | 40 | 167 | 0.19 | 0.03 | 0.33 | 0.20 | 0.02 | 0.36 |
| | 3 | 18 | 2 | 35 | 82 | 31 | 144 | 0.15 | 0.00 | 0.27 | 0.15 | 0.00 | 0.27 |
| | 4 | 16 | 1 | 32 | 102 | 43 | 167 | 0.09 | 0.00 | 0.22 | 0.33 | 0.20 | 0.48 |
| | 5 | 15 | 1 | 31 | 86 | 35 | 147 | 0.12 | 0.00 | 0.26 | 0.17 | 0.03 | 0.30 |
| Site 4 | 1 | 13 | 1 | 27 | 94 | 38 | 157 | | | | | | |
| | 2 | 13 | 2 | 27 | 79 | 34 | 130 | 0.19 | 0.00 | 0.36 | 0.21 | 0.00 | 0.39 |
| | 3 | 12 | 1 | 24 | 58 | 23 | 99 | 0.16 | 0.00 | 0.32 | 0.17 | 0.00 | 0.32 |
| | 4 | 11 | 2 | 23 | 82 | 37 | 129 | 0.10 | 0.00 | 0.25 | 0.42 | 0.26 | 0.62 |
| | 5 | 11 | 0 | 22 | 64 | 24 | 107 | 0.14 | 0.00 | 0.29 | 0.23 | 0.04 | 0.41 |

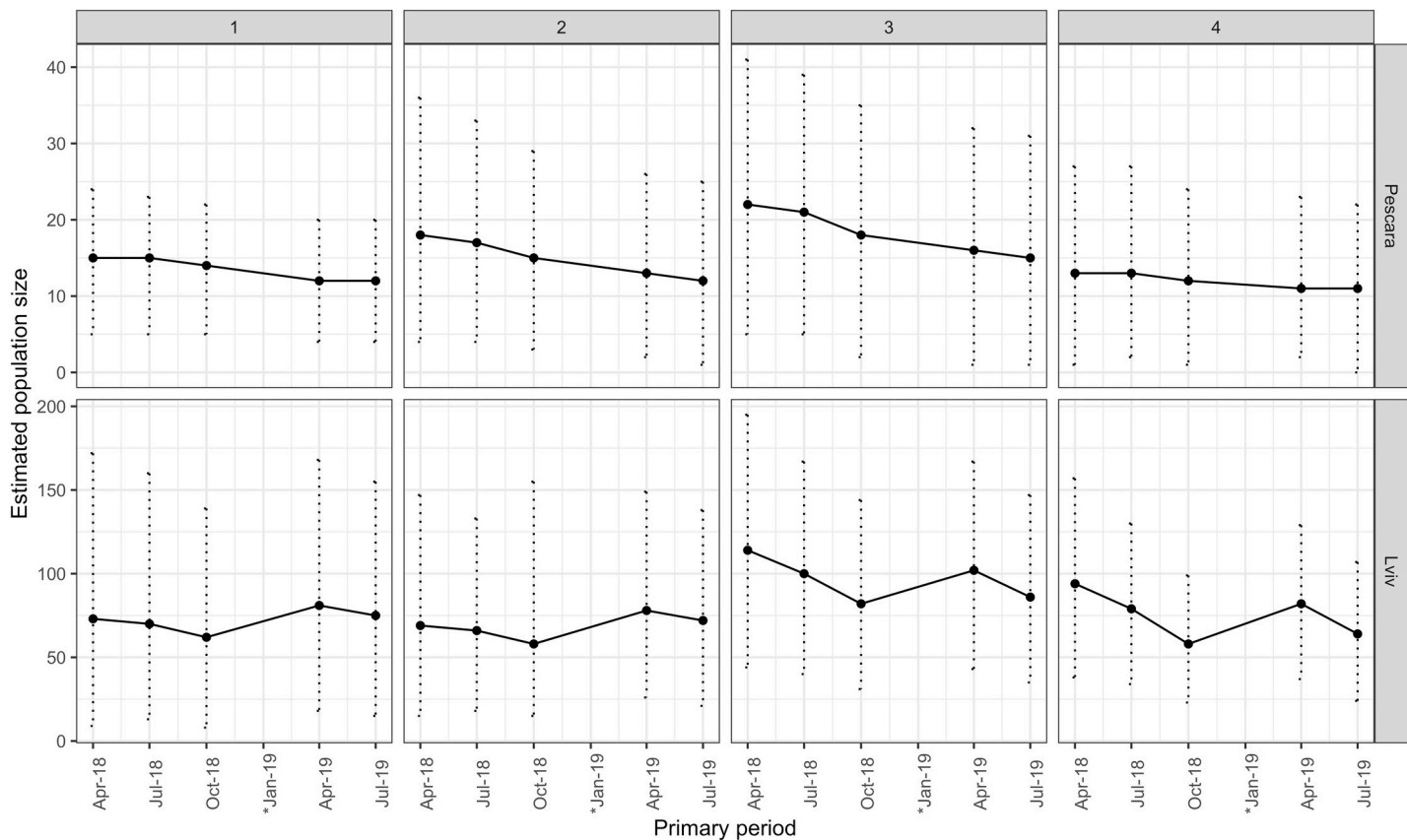

**Fig 4. Estimated population size for each study site (1 to 4) in Pescara, Italy and Lviv, Ukraine across the primary sampling periods between April 2018 and July 2019.** Error bars show the 2.5 and 97.5 percentiles of the posterior distribution (95% CI). *No surveys conducted in January 2019.

There was no significant effect of rain, temperature, sex, study site, or primary period on detection probability in Pescara or Lviv (Table 8). In Lviv, there was a significant effect of weekend on detection probability. When converted to the probability scale, the probability of observing a dog was 0.18 (95% CI: 0.02–0.40) on surveys conducted at the weekend and 0.14 (95% CI: 0.01–0.33) on weekdays (Table 8). There was also a significant effect of market days on detection in Lviv (Table 8). The probability of observing a dog was 0.33 (95% CI: 0.06–0.66) on market days and 0.18 (95% CI: 0.02–0.40) on non-market days.

**Table 8. Effects of predictor variables on detection and apparent survival as odds ratios (OR) in Pescara, Italy and Lviv, Ukraine.** Significant results are highlighted in bold.

| | Detection | | | | | | Apparent survival | | | | | |
|---|---|---|---|---|---|---|---|---|---|---|---|---|
| | Pescara | | | Lviv | | | Pescara | | | Lviv | | |
| | OR | 2.5% CI | 97.5% CI | OR | 2.5% CI | 97.5% CI | OR | 2.5% CI | 97.5% CI | OR | 2.5% CI | 97.5% CI |
| Weekend vs. weekday | 1.16 | 0.64 | 1.74 | **0.74** | **0.53** | **0.96** | | | | | | |
| Market day vs. no market | 0.75 | 0.24 | 1.36 | **2.58** | **1.28** | **4.14** | | | | | | |
| Rain vs. dry | 0.79 | 0.31 | 1.34 | 0.73 | 0.47 | 1.00 | | | | | | |
| Temperature | 0.98 | 0.88 | 1.08 | 0.98 | 0.92 | 1.04 | | | | | | |
| Male vs. female | 0.63 | 0.22 | 1.15 | 0.82 | 0.37 | 1.32 | 1.29 | 0.08 | 3.43 | **0.25** | **0.03** | **0.59** |

There was no evidence for a significant effect of study site or primary period on apparent survival probability in Pescara or Lviv (S8 and S9 Tables). Sex had a significant effect on apparent survival in Lviv (Table 8). When comparing across the average primary period (3–6 months), average male apparent survival probability was 0.40 (95%CI: 0.04–0.76) compared to 0.73 (95%CI: 0.45–0.95) in females. There were no significant differences in detection (Pescara vs Lviv: odds ratio 3.36, 95% CI 0.04 to 10.60) or apparent survival parameters (Pescara vs Lviv: odds ratio 1.57, 95% CI 0.01 to 5.13) between countries.

## Discussion

This study provides some of the first estimates of health, welfare, population size, and rates of recruitment and removal for free-roaming dog populations in Europe. This study is also the first to use Pollock's robust design mark-recapture to estimate free-roaming dog population dynamics. We found high population turnovers in both Pescara and Lviv, with removal rates of 7% per month in both locations, and recruitment rates (per capita entry probabilities) of 5–6% per month. Few juveniles were observed in either location, potentially indicating recruitment through abandonment or immigration. This detailed demographic data provides critical information for planning effective dog population management and informing population and infectious disease modelling.

In Pescara, estimates of dog population size and density were much smaller than in Lviv (7 dogs km$^{-2}$ in Pescara, and 40 dogs km$^{-2}$ in Lviv). As dog population size correlates with human population size [2], the difference in population estimates could relate to the smaller human population sizes in the study villages/towns in Pescara, compared to those in the study sites in the city of Lviv. Population size reduction is an important indicator of effective dog population management. Both Pescara and Lviv had ongoing dog population management through CNR and sheltering. In Pescara, the population size decreased over the study period. Whilst this decreasing trend in dog population size observed in Pescara may relate to the population management, these results should be interpreted with caution, as determining an effect of management would require a baseline period (prior to management intervention), a control population, and an increased number of study sites. As many dog populations have historically been managed to some extent, it is challenging to obtain true baseline or control populations. The length of the study would also need to be increased to distinguish between a reducing population and natural fluctuations in population size, particularly as modelling studies suggest the effect of management may take years to be observed [10,61,62]. Population management through CNR could create more stable populations over shorter periods of time, prior to decreasing the population size, and this would be reflected by low recruitment and removal rates. Both free-roaming dog populations in Pescara and Lviv had high recruitment and removal rates, suggesting further management could be implemented to reduce the size and turnover of these free-roaming dog populations. Recruitment and removal rates provide an alternative indicator of management impact that could be observed over shorter durations and, as such, should be considered in future population monitoring efforts.

There was no evidence for significant effects of study site (within regions) on apparent survival probability. For study sites in Lviv with different levels of management intensity, this possibly indicates that dog population management, as currently applied, does not influence the death rate or migration rate of dog populations. Belo *et al.*, (2017) [34] also found no evidence for a significant effect of management on apparent survival rates. In general, parameter estimates had wide confidence intervals, relating to low sample sizes, limiting the strength of the study's conclusions. Additionally, similar percentages of neutered dogs were observed across all study sites in Lviv (Table 6), even though no management had been recorded in study sites

three and four. This potentially indicates historical dog population management in the area (i.e. CNR conducted by other organisations) or movement of dogs throughout the city. Although we found no evidence of individuals moving between study sites (no individuals were observed in more than one study site), a longer study or increased number of study sites may have captured these movement events. These factors could possibly explain the lack of clear differences between study sites on apparent survival probability. To allow more robust estimates of changes in population size, recruitment, and removal rates and to determine effects of dog population management, future studies should consider (i) conducting mark-recapture over several of years and (ii) having a larger sample size in terms of study sites and secondary sampling periods.

The observed sex ratio, percentage of adults, and body condition scores were similar between Pescara and Lviv, and we found no evidence of country effects on detection or apparent survival probability, despite differences in habitats (rural vs. urban), climates, and culture. The average monthly apparent survival probabilities (i.e. the probability of an individual surviving and remaining in the study site) of 93% in Pescara and Lviv are similar to those reported by Belo et al., (2017) [34] in Brazil of between 86–99% per month using the Jolly-Seber open mark-recapture approach. The reported per capita entry probabilities (i.e. fraction of all dogs in the population entering at each time point) of 5–6% per month in both Pescara and Lviv (Table 7) are also similar to those reported by Belo et al., (2017) [34] of 0–8% per month for free-roaming dog populations in Brazil. Similar rates of recruitment and removal may suggest similar birth, mortality, and movement rates of free-roaming dogs between the different countries. To determine this, future work should aim to disentangle the processes of recruitment and removal to provide rates of birth, mortality, abandonment, emigration, and immigration (both to neighbouring populations and from different subpopulations) within the free-roaming dog population. Disentangling the processes of recruitment and removal is a challenging task. For example, while mark-recapture methods incorporating age cohorts have been used for other species to disentangle births from immigration [63], in free-roaming dog populations these methods would not be of use, as dogs may be abandoned at any age. Instead, disentangling these processes may require working with local communities to assist in these estimates, for example through questionnaire surveys or interviews with focus groups.

High recruitment and removal rates, and therefore high population turnover, can hinder population and infectious disease control that aims to maintain a neutered/vaccinated coverage above a critical threshold. For example, a primary motive for dog population management globally is to reduce the risk of rabies transmission from dogs to humans [64]. Vaccination of 70% of the dog population is required to reduce transmission or prevent an outbreak of rabies in a population [65]. High death rates and movement of vaccinated dogs out of an area, and high birth rates and movement of susceptible, unvaccinated dogs into an area may reduce the overall coverage to below this critical threshold. Rabies is a public health concern in Ukraine, with outbreaks occurring sporadically in free-roaming dog populations [42,66]. We observed vaccination and neutering coverages of only 17–52% across the study sites in Lviv, which are insufficient to prevent an outbreak of rabies in these populations. Removal of 7% and recruitment of 5–6% of the population per month has the potential to further reduce vaccination and neutering coverages. Higher vaccination coverages and reduced population turnover are required to protect against a rabies outbreak in Lviv.

Our findings suggest that 7% of the population per month is removed through deaths, or by movement to other populations, such as adoption to the restricted owned dog population, or migration to another section of the city. The recorded lifespan of free-roaming dogs is low, often reported as under three years for populations in Africa and Asia [17,21,67], although estimates are lacking for free-roaming dog populations in Europe. A lifespan of three years

translates to a mortality rate of approximately 3% per month, suggesting that movement accounted for much of the removal rate observed in this study.

We observed recruitment rates (per capita entry probabilities) of 5–6% per month. It is challenging to disentangle whether these individuals were recruited through births, abandonment, or immigration. Throughout the study, and in both study regions, few juveniles and lactating females and no visibly pregnant females were observed, suggesting low birth rates with recruitment instead occurring through movement from other populations, such as immigration or abandonment of adult dogs. Although it is important to consider that fewer juveniles may have been observed due to a possible lower detection probability, for example, if puppies were hidden with their mother, out of sight of the observer, in dens, bushes or under cars. Belo et al., (2017) [34] in Brazil and Morters et al., (2014) [16] in Indonesia and South Africa also determine that recruitment was primarily driven by the movement of adult dogs in their study populations. This has important implications for population control. If dogs are recruited through abandonment, management efforts should be targeted at responsible dog ownership to reduce abandonment and the prevalence of free-roaming owned dogs, particularly those that are intact. Similarly, to mitigate the effects of immigration of intact dogs between sections of a city, interventions should be planned to ensure whole-city coverage, as movement of dogs between sections of the city could quickly repopulate areas, reaching carrying capacity through either births or migration.

In Lviv, males had a lower apparent survival probability (0.40 compared to 0.73 in females). As other studies report higher mortality rates in female dogs [17,18,68], it is likely that the lower apparent survival probabilities in males is due to increased movement. Movement of individuals is related to resources, and, for males, these resources may include seeking mates, possibly resulting in increased migration and lower apparent survival probabilities compared to females. This is supported by studies investigating home range sizes [69] and dispersal behaviour [70] of free-roaming dogs, that find greater dispersal and movement in intact males, compared to females or neutered males. Movement of free-roaming dogs may reduce local vaccination or neutering coverage, which can hinder disease and population management. As intact males are more likely to disperse than neutered males, targeting the neutering of male dogs may help reduce population turnover and maintain high local vaccination/neutering coverages.

In the study regions, the average detection probability was slightly lower than those reported in other studies of free-roaming dogs (Pescara 0.27, 95% CI 0.05 to 0.54; Lviv 0.18, 95% CI 0.02 to 0.40) (Table 5), which range between 0.33 and 0.68 for dog populations in Brazil and India [34,46,71,72]. Detection probability is dependent upon an individual being present in the study area, available for detection, and detected during mark-recapture surveys [73]. The slightly lower detection probability reported in this study could possibly be due to differences in the structure of the study areas, in human-dog interactions, or in the mark-recapture models used to estimate this parameter [46].

Detection probabilities were higher for surveys conducted on the weekend in Lviv, although not in Pescara. The lack of evidence of effect on detection probability in Pescara may be due to the smaller number of observed dogs (Pescara 53; Lviv 182), leading to smaller sample sizes to determine an effect. Several studies describe differences in dog detection probability due to time of day effects, for example higher detection probabilities in the morning compared to the afternoon [44–47]. This study highlights the importance of future studies also considering potential day of the week influences. This effect may relate to changes in human behaviour and activity at the weekend compared to on weekdays. For example, there may be a reduction in human activity due to fewer people travelling for work at the weekend, potentially leading to higher free-roaming dog activity, and therefore detectability, when streets are quieter. Similarly, we found a significant effect of market events on detection probability in Lviv. This may

again relate to human activity and behaviour, such as high aggregations of people and potential food resources. Tiwari *et al.* (2018) [46] also found higher detection rates related to human events. Human activity and behaviour (for example, due to events or public holidays) need to be considered in mark-recapture analyses, particularly when interpreting results across time or areas.

In this study, a photographic method was used to identify individuals, limiting the impact of the "marking" on detection probability. Photographic methods are advantageous over other methods used to mark dogs, such as dyes that require animal contact [72]. In this study, all individuals were assumed to be correctly re-identified. However, errors in capture histories could have occurred, particularly for less-distinct individuals. These errors can lead to less accurate parameter estimates. Most dogs were classified as very or moderately distinct (88% in Pescara; 82% in Lviv) and were more likely to be correctly re-identified. It is worth noting that higher percentages of indistinct individuals may occur in free-roaming dog populations in other geographic areas. The applicability of photographic mark-recapture methods may be limited in populations with high proportions of indistinct individuals. For these populations, use of tags (such as ear tags) or other long-term individually identifiable markings could be used but are less advantageous as the use of tags often requires capture and handling to read individual identifiers. Additionally, photographic mark-recapture studies may benefit from photograph matching software to reduce error rates and increase accuracy of parameter estimates [74].

## Conclusions

This study has provided detailed dog population dynamics data that is critical for informing population and infectious disease models and planning effective control strategies. We found high population turnovers in both Pescara and Lviv, with removal rates of 7% per month in both locations, and recruitment rates (per capita entry probabilities) of 5–6% per month. Few juveniles were observed in this study, potentially providing evidence for recruitment and removal through movement between dog subpopulations. Future management should be conducted to ensure entire municipality coverage and incorporate management of owned unrestricted dog populations (preventing reproduction through restricted movement or reproductive control). This study has also identified that detection probability of dogs may be influenced by day of the week, and human events, such as markets. Future researchers conducting mark-recapture of free-roaming dog populations should consider controlling for these effects–statistically or through study design–to ensure surveys are comparable across time and between areas.

## Supporting information

**S1 Fig. Posterior distribution of estimated population size (N) at primary sampling period 1 in study sites in Pescara and Lviv.**
(TIF)

**S2 Fig. Posterior distribution of estimated population size (N) at primary sampling period 2 in study sites in Pescara and Lviv.**
(TIF)

**S3 Fig. Posterior distribution of estimated population size (N) at primary sampling period 3 in study sites in Pescara and Lviv.**
(TIF)

**S4 Fig. Posterior distribution of estimated population size (N) at primary sampling period 4 in study sites in Pescara and Lviv.**
(TIF)

**S5 Fig. Posterior distribution of estimated population size (N) at primary sampling period 5 in study sites in Pescara and Lviv.**
(TIF)

**S6 Fig. Population growth rates between primary sampling periods in study sites 1 to 4 for study regions Pescara, Italy.** Error bars show the 2.5 and 97.5% limits of the highest posterior density credible intervals (CI) of the posterior distribution. Blue lines indicate stable population (i.e. no growth or decline). *Note uneven spacing as no surveys conducted in January 2019.
(TIF)

**S7 Fig. Population growth rates between primary sampling periods in study sites 1 to 4 for study regions Lviv, Ukraine.** Error bars show the 2.5 and 97.5% limits of the highest posterior density credible intervals (CI) of the posterior distribution. Blue lines indicate stable population (i.e. no growth or decline).* Note uneven spacing as no surveys conducted in January 2019.
(TIF)

**S1 Table. Numbers of dogs caught, neutered and released to study sites in Pescara, Italy and Lviv, Ukraine between 2014 and 2019.** Sources: Veterinary Services–Pescara Province Local Health Unit for Pescara; and local Communal Enterprise for Lviv.
(DOCX)

**S2 Table. Survey timings, distance and length (minimum, maximum and mean) in study sites in Pescara, Italy and Lviv, Ukraine.**
(DOCX)

**S3 Table. Primary and secondary sampling period timings, temperature and weather conditions in Pescara, Italy.**
(DOCX)

**S4 Table. Primary and secondary sampling period timings, temperature and weather conditions in Lviv, Ukraine.**
(DOCX)

**S5 Table. Probability of apparent survival and detection for primary sampling periods (averaged across individuals and study sites) and study sites (averaged across individuals and primary periods) in Pescara, Italy.**
(DOCX)

**S6 Table. Probability of apparent survival and detection for primary sampling periods (averaged across individuals and study sites) and study sites (averaged across individuals and primary periods) in Lviv, Ukraine.**
(DOCX)

**S7 Table. Standard deviations for between-dog effects on survival and detection on log odds scale.**
(DOCX)

**S8 Table. Comparison of mean apparent survival and detection as odds ratios between different study sites in Pescara, Italy and Lviv, Ukraine.**
(DOCX)

**S9 Table. Comparison of mean apparent survival and detection as odds ratios between different intervals between primary periods in Pescara, Italy and Lviv, Ukraine.**
(DOCX)

**S1 File. Survey timings.**
(DOCX)

**S2 File. Details of hierarchical Bayesian hidden Markov model of Pollock's robust design.**
(DOCX)

## Acknowledgments

We thank Sarah Ross and Alesya Lischyshyna for project and logistical support; Marta Kopach, Viktor Kopach, Vasil Dub and all at Animal-id.info for providing technology to aid in data collection; all field assistants who helped to collect the data–Dr Helen Gray, Dr Mary Friel, Annika Geijer-Simpson, Phoebe Abrahams, and Sarah Ross; the local Communal Enterprise in Lviv, and the Veterinary Services–Pescara Province Local Health Unit who provided data on dog population management in the study areas.

## Author Contributions

**Conceptualization:** Lauren Margaret Smith, Rupert J. Quinnell, Alexandru M. Munteanu, Sabine Hartmann, Paolo Dalla Villa, Lisa M. Collins.

**Data curation:** Lauren Margaret Smith.

**Formal analysis:** Lauren Margaret Smith, Conor Goold, Rupert J. Quinnell.

**Funding acquisition:** Lisa M. Collins.

**Investigation:** Lauren Margaret Smith.

**Methodology:** Lauren Margaret Smith, Conor Goold, Rupert J. Quinnell, Lisa M. Collins.

**Supervision:** Rupert J. Quinnell, Lisa M. Collins.

**Writing – original draft:** Lauren Margaret Smith.

**Writing – review & editing:** Lauren Margaret Smith, Conor Goold, Rupert J. Quinnell, Alexandru M. Munteanu, Sabine Hartmann, Paolo Dalla Villa, Lisa M. Collins.

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
