## [Decision Letter · Decision Letter 0]

12 Jul 2022

PONE-D-22-07758Population dynamics of free-roaming dogs in Europe and implications for population controlPLOS ONE

Dear Dr. Smith,

Thank you for submitting your manuscript to PLOS ONE. After careful consideration, we feel that it has merit but does not fully meet PLOS ONE’s publication criteria as it currently stands. Therefore, we invite you to submit a revised version of the manuscript that addresses the points raised during the review process.

I first would like to apologize to the authors for the delay in returning a decision on this manuscript. It has been a challenging time to find reviewers, and your paper has now been reviewed by one expert and myself. The reviewer offers many suggestions for improvement, including methodological suggestions, clarifying points in the framing, and suggestions on result inclusion/exclusion.==============================

We look forward to receiving your revised manuscript.

Kind regards,

Daniel Becker

Academic Editor

PLOS ONE

Journal Requirements:

2. In your Methods section, please provide additional information regarding the permits you obtained for the work. Please ensure you have included the full name of the authority that approved the field site access and, if no permits were required, a brief statement explaining why

Reviewers' comments:

Reviewer's Responses to Questions

**Comments to the Author**

1. Is the manuscript technically sound, and do the data support the conclusions?

Reviewer #1:Yes

2. Has the statistical analysis been performed appropriately and rigorously? 

Reviewer #1:Yes

3. Have the authors made all data underlying the findings in their manuscript fully available?

Reviewer #1:Yes

4. Is the manuscript presented in an intelligible fashion and written in standard English?

Reviewer #1:Yes

5. Review Comments to the Author

Reviewer #1:I revised the manuscript entitled "Population dynamics of free-roaming dogs in Europe and implications for population control" by Smith and colleagues.

The topic of this study is of great interest and the results are useful.

Conversely, some points need to be addressed properly to improve the quality of the manuscript as suggested below.

-The title does not adequately represent the study. Results are not applicable to Europe as a whole, but to specific regions of two countries.

-The conclusions presented in lines 35 - 37 bear little relation to the results of the study.

-What aspects of the studied areas make the study relevant?

Can Pescara and Lviv represent other regions? What is the external validity of the data?

What is the relevance of unrestricted dogs in the local context? Are there health, environmental, safety and animal welfare issues related to dogs in these areas? What are the prevalence and incidence of zoonoses?

-The article presents a multitude of supplementary materials. Because they present essential information, I suggest that the following materials be incorporated into the text:

S1 Fig; S1 file; S2 file; S4 file; S5 table.

-It is also important that all remaining supplementary materials are properly mentioned in the article. In other words, specific mentions should be made to each one of them.

-Consider the inclusion of a map of Italy and Ukraine, Pescara and Lviv, and study areas, to facilitate the reader to locate the sites of study.

-There is a major contradiction in the explanation presented between lines 157-161 and in the S4 File. In which primary and secondary samplings, in fact, there were no collections?

How might this have affected the results?

-Explain what it is and how registrations were made in the Animal-id.info app (animal-id.net).

-Since the term "robust design" is cited in the introduction and discussion, the analysis explanation (starting at line 186) must also use it (at some point).

-The logic behind the choice of the hierarchical Bayesian Markov model must be explained. Relate to other possible models within the context of robust design and justify its adoption.

-How were the models built and compared? How was the goodness of fit assessed? What results were obtained?

-Important descriptive information is missing:

number of dogs identified in each sample;

number of new dogs in each sampling;

number of dogs in each collection that had already been identified in previous samplings.

-Estimates of abundance/population growth should be better presented and discussed.

-The authors describe that:

"Few juveniles were observed in either location, indicating recruitment through abandonment or immigration.".

While this is plausible, it should be considered that the "probability of captures" of younger dogs may be lower.

-The authors describe that:

"Similar rates of recruitment and removal may suggest similar birth, mortality, and movement rates of free-roaming dogs between the different countries"

Considering that the areas being compared have quite different socioeconomic and cultural characteristics, this statement should be better addressed. In addition, it is important to point out that the issue should be further investigated in the literature.

6. PLOS authors have the option to publish the peer review history of their article (what does this mean?). If published, this will include your full peer review and any attached files.

Reviewer #1:No

While revising your submission, please upload your figure files to the Preflight Analysis and Conversion Engine (PACE) digital diagnostic tool,https://pacev2.apexcovantage.com/. PACE helps ensure that figures meet PLOS requirements. To use PACE, you must first register as a user. Registration is free. Then, login and navigate to the UPLOAD tab, where you will find detailed instructions on how to use the tool. If you encounter any issues or have any questions when using PACE, please email PLOS atfigures@plos.org. Please note that Supporting Information files do not need this step.

---

## [Author Response · Author response to Decision Letter 0]

4 Aug 2022

Dear Editor,

We would like to thank both you and the reviewer for your valuable feedback on our manuscript. We have addressed each of the comments, as outlined below and updated the manuscript using track changes.

Reviewer 1

Reviewer #1: I revised the manuscript entitled "Population dynamics of free-roaming dogs in Europe and implications for population control" by Smith and colleagues.

The topic of this study is of great interest and the results are useful.

Conversely, some points need to be addressed properly to improve the quality of the manuscript as suggested below.

-The title does not adequately represent the study. Results are not applicable to Europe as a whole, but to specific regions of two countries.

Thank you for your feedback – we’ve updated the title to: “Population dynamics of free-roaming dogs in two European regions and implications for population control”.

-The conclusions presented in lines 35 - 37 bear little relation to the results of the study.

We have now omitted these conclusions.

-What aspects of the studied areas make the study relevant?

Can Pescara and Lviv represent other regions? What is the external validity of the data?

What is the relevance of unrestricted dogs in the local context? Are there health, environmental, safety and animal welfare issues related to dogs in these areas? What are the prevalence and incidence of zoonoses?

We have now added detail of the free-roaming dog populations and their risks in the study regions, as well as the representativeness of Pescara and Lviv to other regions (Lines 121-128). 

-The article presents a multitude of supplementary materials. Because they present essential information, I suggest that the following materials be incorporated into the text:

S1 Fig; S1 file; S2 file; S4 file; S5 table..

Thank you for this suggestion, we have now included the above materials into the text.

-It is also important that all remaining supplementary materials are properly mentioned in the article. In other words, specific mentions should be made to each one of them.

We have now ensured all supplementary material are referred to properly within the manuscript.

-Consider the inclusion of a map of Italy and Ukraine, Pescara and Lviv, and study areas, to facilitate the reader to locate the sites of study.

We have included a figure (figure 1) highlighting the countries of Italy and Ukraine and indicating the study regions of Pescara and Lviv. A detailed map of the precise study sites could not be included, as these remain anonymous as a condition of data sharing. 

-There is a major contradiction in the explanation presented between lines 157-161 and in the S4 File. In which primary and secondary samplings, in fact, there were no collections?

How might this have affected the results?

Data was collected for every scheduled primary period, apart from one secondary sampling period (one survey) in study site one in Lviv in October 2019. We missed one survey in this one study site as I was unwell and could not conduct the survey on this day. Due to logistics, we were unable to extend our fieldwork and include an extra day of data collection to make up the lost secondary sampling period. This does not impact the results, as the data is “missing completely at random”, i.e. the probability of missing data is unrelated to the processes of data observation. We have amended the text to describe this missing survey day in one study site more accurately and signpost the reader to the supplementary information (lines 181-184). We also include how we dealt with this missing information in the model in lines 330-334.

-Explain what it is and how registrations were made in the Animal-id.info app (animal-id.net).

We have added more explanation of this app (lines 201-203).

-Since the term "robust design" is cited in the introduction and discussion, the analysis explanation (starting at line 186) must also use it (at some point).

We have now amended to describe that the model is “a hierarchical Bayesian hidden Markov model of Pollock’s robust design.

-The logic behind the choice of the hierarchical Bayesian Markov model must be explained. Relate to other possible models within the context of robust design and justify its adoption.

We have added description of the logic and advantages of the hierarchical Bayesian hidden Markov model (Lines 233-242).

-How were the models built and compared? How was the goodness of fit assessed? What results were obtained?

We have added discussion in the text to explain our reasoning for not conducting an explicit model comparison in this study:

“It is common in frequentist mark-recapture modelling to run several models, stratified by temporal and population subgroup parameter estimates (leading to an enormous number of possible models, see [58] for discussion) and to use model-averaging to provide parameter results. Studies suggest the use of Hierarchical Bayesian mark-recapture models with random-effects yield similar parameter estimates to model-averaging of frequentist mark-recapture models (using Akaike Information Criterion, AICc weights) [58–60]. As all parameters in the model are of theoretical relevance and as there were no specific biological hypotheses, no explicit model comparison was run.” Parameters in the model experienced both shrinkage effects (for the random effects describing differences between individuals, time points and study sites), and weakly informative priors were used to exclude large effects. We assessed model convergence by the Rhat and effective sample size values. 

-Important descriptive information is missing:

number of dogs identified in each sample;

number of new dogs in each sampling;

number of dogs in each collection that had already been identified in previous samplings.

We have added in a new table (Table 4) and have included this information in the text (Lines 348-353).

-Estimates of abundance/population growth should be better presented and discussed.

We have now included the estimated population sizes for each study site and primary period in Table 7. We discuss the issues with over-interpreting the changes in population size in this study in lines 454-475, and add some further discussion to this section.

-The authors describe that:

"Few juveniles were observed in either location, indicating recruitment through abandonment or immigration.".

While this is plausible, it should be considered that the "probability of captures" of younger dogs may be lower.

Thank you for highlighting this – we agree and have added in lines 449 and 618 “potentially” to be more tentative with our suggestion, and have added “Although it is important to consider that fewer juveniles may have been observed due to a possible lower detection probability, for example, if puppies were hidden with their mother, out of sight of the observer, in dens, bushes or under cars.” in lines 545-548.

-The authors describe that:

"Similar rates of recruitment and removal may suggest similar birth, mortality, and movement rates of free-roaming dogs between the different countries"

Considering that the areas being compared have quite different socioeconomic and cultural characteristics, this statement should be better addressed. In addition, it is important to point out that the issue should be further investigated in the literature.

Interestingly, we found no evidence of country level effects on detection or apparent survival probability, despite the different socioeconomic and cultural characteristics. We also found that apparent survival and entry probabilities were similar to those reported in Brazil. We have added a section to emphasise the importance of disentangling the processes of recruitment and removal in lines 507-515.

After consultation with our partners within Italy and Ukraine, we determined that no permits were required for this study. All data was collected in public areas of Italy and Ukraine (i.e., publicly accessible streets). We have added this detail in the methods section (Lines 157-159).

---

## [Decision Letter · Decision Letter 1]

24 Aug 2022

Population dynamics of free-roaming dogs in two European regions and implications for population control

PONE-D-22-07758R1

Dear Dr. Smith,

We’re pleased to inform you that your manuscript has been judged scientifically suitable for publication and will be formally accepted for publication once it meets all outstanding technical requirements.

Kind regards,

Daniel Becker

Academic Editor

PLOS ONE

Additional Editor Comments (optional):

Reviewers' comments:

Reviewer's Responses to Questions

**Comments to the Author**

1. If the authors have adequately addressed your comments raised in a previous round of review and you feel that this manuscript is now acceptable for publication, you may indicate that here to bypass the “Comments to the Author” section, enter your conflict of interest statement in the “Confidential to Editor” section, and submit your "Accept" recommendation.

Reviewer #1:All comments have been addressed

2. Is the manuscript technically sound, and do the data support the conclusions?

Reviewer #1:Yes

3. Has the statistical analysis been performed appropriately and rigorously? 

Reviewer #1:Yes

4. Have the authors made all data underlying the findings in their manuscript fully available?

Reviewer #1:Yes

5. Is the manuscript presented in an intelligible fashion and written in standard English?

Reviewer #1:Yes

6. Review Comments to the Author

Reviewer #1:(No Response)

7. PLOS authors have the option to publish the peer review history of their article (what does this mean?). If published, this will include your full peer review and any attached files.

Reviewer #1:**Yes:**Vinícius Silva Belo

---

## [Editor Report · Acceptance letter]

30 Aug 2022

PONE-D-22-07758R1 

Population dynamics of free-roaming dogs in two European regions and implications for population control 

Dear Dr. Smith:

I'm pleased to inform you that your manuscript has been deemed suitable for publication in PLOS ONE. Congratulations! Your manuscript is now with our production department. 

Kind regards, 

on behalf of

Dr. Daniel Becker 

Academic Editor

PLOS ONE